# Varied and unexpected changes in the well-being of seniors in the United States amid the COVID-19 pandemic

Silvia Barcellos[1,2☉], Mireille Jacobson[iD][3,4☉]*, Arthur A. Stone[1,2,4,5,6☉]

1 Center for Economic and Social Research (CESR), University of Southern California, Los Angeles, CA, United States of America, 2 Department of Economics, University of Southern California, Los Angeles, CA, United States of America, 3 Davis School of Gerontology, University of Southern California, Los Angeles, CA, United States of America, 4 Schaeffer Center for Health Policy & Economics, University of Southern California, Los Angeles, CA, United States of America, 5 Dornsife Center for Self-Report Science, University of Southern California, Los Angeles, CA, United States of America, 6 Department of Psychology, University of Southern California, Los Angeles, CA, United States of America

☉ These authors contributed equally to this work.
* mireillj@usc.edu

**Data Availability Statement:** All relevant data are within the paper and its S1–S6 Files, S1&S5 Tables.

**Funding:** SB received support from National Institutes on Aging grants NIA K01AG050811-

## Abstract

Recent evidence suggests that psychological health deteriorated during the COVID-19 pandemic but far less is known about changes in other measures of well-being. We examined changes in a broad set of measures of well-being among seniors just before and after the recognition of community spread of COVID-19 in the United States. We fielded two waves of a survey to a large, national online panel of adults ages 60 to 68 at wave 1. We measured depressive symptoms, negative affect, positive affect, pain, life satisfaction and self-rated health in each survey wave. 16,644 adults answered well-being questions in waves 1 and 2 of our survey (mean[SD]: age 64 [2.6]; 10,165 women [61%]; 15,161 [91%] white). We found large (20%; p<0.001) increases in the rate of depressive symptoms (1.4 percentage points; 95% CI, 0.97 to 1.86) and negative mood (0.225 scale points; 95% CI, 0.205 to 0.245) but no change in self-reported health and a decrease (12.5%; p<0.001) in the rate of self-reported pain (5 percentage points; 95% CI, -5.8 to -4.3). Depressive symptoms and negative affect increased more for women. Higher perceived risk of getting COVID-19 and of dying from the disease were associated with larger increases in the rate of depressive symptoms and negative affect and larger decreases in positive affect and life satisfaction. COVID-19 related job/income loss was the only pandemic-related factor predictive of the decline in pain. Although depressive symptoms and mood worsened during the COVID-19 pandemic, other measures of well-being were either not materially affected or even improved.

03S1 and NIA K01AG050811-05S1 (https://www.nia.nih.gov/). MJ received support from Navigage Foundation (https://navigage.org/).

**Competing interests:** The authors have declared that no competing interests exist.

## Introduction

The novel coronavirus (COVID-19) upended the lives of many individuals around the world. On March 11, 2020, the World Health Organization declared the outbreak a global pandemic and two days later the United States declared a national emergency. Shortly thereafter 42 U.S. states and territories issued mandatory stay-at-home orders, placed restrictions on gatherings and non-essential businesses, and ordered school closures [1, 2]. Work moved to home, when feasible; nursing homes and assisted living facilities barred visitors; nuclear families isolated, limiting interaction with friends, grandparents, and other relatives; and mobility plummeted, even in places without restrictions.[2] COVID-19 cases, hospitalizations, and deaths soared. While confined initially to the Northeast, Louisiana and a handful of other states, COVID-19 morbidity and mortality ultimately affected every part of the country [3]. In short succession, as a result of policies, precautions, and in some cases fear, life in the U.S. changed dramatically.

Multiple studies highlight the toll the pandemic has taken on the mental health of Americans. The prevalence of depressive symptoms among US adults increased from about 11% in March 2020 to 14% in April 2020 [4]. A comparison of severe psychological distress among US adults ages 18 and over in April 2020 relative to 2018 found that prevalence had increased nearly three-fold [5]. Another study found increases in all categories of depressive symptoms after the pandemic [6]. Mental distress was higher in areas with more COVID-19 cases and among individuals who perceive the risks of infection and death from the disease as higher [7]. A worsening of mental health has been documented in both parents and children [8, 9] and has been shown to be larger among vulnerable sub-groups [10]. Multiple studies, both domestic and international, have found that seniors have fared better than other age groups during pandemic, although the reasons are unclear and potentially complex [11].

There are, however, some concerns about these findings. Most of these studies are either cross-sectional [4–6, 12–14] or follow respondents post-pandemic [10, 15], with few exceptions [7, 16]. Thus, they cannot evaluate changes in well-being before and after the start of the pandemic. Study samples tend to be small and generally do not assess how the pandemic has affected measures of well-being other than depressive symptoms or anxiety and how such effects relate to individual-level characterstics and circumstances.

To address these concerns, we conducted analyses of two surveys of a large cohort of seniors that were conducted just before and after the recognition of community spread of COVID-19 in the US (S1 and S2 Files). Having "pre-COVID" data is uncommon in previous work, which typically uses data from another study sample as a basis for comparison. By following the same people over time, we can estimate within person changes in well-being and investigate individual-level sources of heterogeneity in such changes.

Our survey captured a broad set of measures of well-being including not only depressive symptoms, but also negative affect, positive affect, life satisfaction, self-rated health, and pain. We analyzed changes in these measures across survey waves. We hypothesized that well-being would: decline across a range of dimensions, not just mental health, between wave 1 and 2 of our survey; that these changes would vary by baseline characteristics such as gender and education; and that they would be related to the severity of the local COVID-19 pandemic, personal economic circumstances, and perceptions of COVID-19 risk. Our analysis of heterogeneity was exploratory in nature given the uncertain effects of a rare event such as a pandemic. However, the analysis was motivated by the potential for COVID-19 to differentially affect individuals based on circumstances such as caregiver status, work status, ability to work from home, and financial exposure, which are likely related to education, gender, race and other "predetermined" characteristics in our survey.

## Methods

Between November 2019 and February 2020, we conducted an internet survey of 26,146 respondents living in the US between the ages of 60 and 68. The original research intended to investigate the relationship between Medicare eligibility and well being, which accounts for the age range. The survey instrument included validated measures of mental health and subjective well-being. Respondents were drawn from a U.S. national opt-in internet panel run by Dynata Corporation and comprising about one million households [17]. It is used increasingly for research purposes, including COVID-19 research [18]. Because the panel is proprietary and researchers are only given data from completed surveys, we cannot calculate survey response rates for wave 1.

For the purpose of investigating the impact of the COVID-19 situation, beginning in the second week of April through late May 2020, we re-contacted all participants from our baseline survey and invited them to a follow-up survey that included our original questions and questions about perceptions of COVID-19 risk and COVID-19 related changes in behavior. We had a 79% response rate, with 66% of the original wave 1 sample answering all questions in the follow-up survey. Our analytic dataset included the 16,644 respondents who answered all questions in both surveys. Respondents who did not answer wave 2 were more likely to be female, minority, unmarried, to have been diagnosed with depression and to have lower income in wave 1 (see S1 Table). We matched respondents based on their county of residence to the cumulative COVID-19 county death count at wave 2 as of the day they answer wave 2 and county population. Death data were taken from the Center for Systems Science and Engineering (CSSE) at Johns Hopkins University, which aggregates data from state and local departments of health. County population estimates are from the 2018 American Community Survey 1-Year Data estimates. In our analyses, we consider death rates, a county's cumulative COVID-19 death count divided by the county population, to account for the wide variation in county size.

### Ethics statement

The University of Southern California's Instiutional Review Board approved and deemed this study human subjects exempt (UP-20-00259) because the investigators do not obtain or record any information that can be readily used to ascertain the identity of the human subjects and disclosure of the responses outside of the research would not reasonably place the subjects at risk. Informed consent of survey respondents is obtained by the data vendor, Dynata. Survey respondents were also shown a survey-specific information sheet before opting into the study. An analytic dataset is available as S5 File.

### Measures

We analyzed 6 measures of well-being: (1) depressive symptoms, (2) negative affect, (3) positive affect, (4) pain, (5) Cantril ladder, (6) self-rated health. Depressive symptoms were captured using the 2-item version of the Patient Health Questionnaire (PHQ-2) that asks separately about how often an individual has experienced the primary symptoms of depression: dysphoric mood (feeling, "down, depressed or hopeless") and anhedona ("little interest or pleasure in doing things" over the past two weeks) [19]. Each of the two questions is scored 0–3 (based on answers ranging from "not at all" to "nearly every day") and scores are summed. Those with a PHQ-2 score of 3 or above were coded as having depressive symptoms, which has been previously shown to have very good sensitivity and reasonable specificity for detecting depressive disorders [20]. Negative affect, positive affect and pain were measured with a set of questions used in the Gallup Health and Wellbeing Index that ask separately about whether

respondents experienced the following feelings "a lot of the day yesterday": enjoyment, happiness, physical pain, worry, sadness, stress, and anger. The order of appearance of each of these feelings was randomized across respondents to avoid any systematic priming. Negative affect is the sum of responses about feelings of worry, sadness, stress and anger and varies from 0 to 4; positive affect is the sum of responses about feelings of enjoyment and happiness and varies from 0 to 2. These measures have been used previously by economists, psychologists, and other behavioral scientists, including several by us [21–23]. We also used the Cantril Self-Anchoring Striving Scale, which has been used widely to measure "judgments of life" or "life evaluation" in contrast to affect [24]. In the version we use, we ask respondents "On which step of the ladder would you say you personally feel you stand at this time?" We use the standard Cantril anchors such that the top of the ladder (scored a ten) represents "the best possible life" and the bottom of the ladder (scored a 0) represents "the worst possible life" [25]. Self-rated health is measured by the standard 5-point scale varying from excellent (1) to poor health (5).

## Statistical analyses

To benchmark our data, we compared our baseline survey to the 2020 Annual Social and Economic Supplement (ASEC) of the Current Population Survey (CPS) for the US population between the ages of 60 to 68. In addition to geographic and demographic information, the ASEC captured self-rated health. We use the 2018 Medical Expenditure Panel Survey (MEPS), the most recent publicly available version of the data, for respondents ages 60 to 68 to compare depressive symptoms.

We compared mean outcomes across waves and performed multivariate regression analysis of changes in outcomes across waves. Our primary specification was

$$WB_{it} = \alpha_0 + \alpha_1 wave2 + \mu_i + \varepsilon_{it} \tag{1}$$

where $WB_{it}$ is a well-being measure, such as positive affect, for individual in $i$ wave $t$ (i.e., wave 1 or 2), $wave2$ is an indicator for wave 2 and $\mu_i$ is an individual fixed effect. The individual fixed effect captures any characteristics, such as race, gender, childhood experiences and so on, that are constant across survey waves. We also conducted stratified regressions by key demographic characteristics of interest–gender, education, race, marriage status, wave 1 retirement status, and income below $50,000 at wave 1. Standard errors were clustered by individual to account for repeated (wave 1 and wave 2) measures.

We also investigated the relationship between across-wave changes in well-being and COVID-19. Specifically, we estimated the following model of the Z-score of the change in well-being across waves:

$$Z_{(WB_{i2} - WB_{i1})} = \alpha_0 + COVID_2'\Theta + X'\beta + \varepsilon_{i2} \tag{2}$$

where $COVID$ is a vector containing (i) an indicator for whether the respondent lived in a 90th percentile COVID-19 death rate county, (ii) an indicator for whether the respondent reported losing his/her job or a substantial portion of income as a result of COVID-19, (iii) the respondent's best guess of the odds he/she 1) would get COVID-19 and, separately, 2) die from the disease if he/she contracted it, and (iv) an indicator for whether COVID-19 kept the respondent from exercising. The vector X captured gender, education, race, marriage status, wave 1 retirement status, and household income below $50,000 at wave 1. All variables were dichotomized so that estimates are relative to the omitted category (e.g. married versus unmaried). We analyzed the Z-score instead of the raw changes in order to compare the relative impacts of each of these factors across the changes in different measures of well-being. Standard errors

were clustered by county to account for repeated spatial correlation in our COVID-19 measures. All analyses were performed using Stata 15.

## Results

Table 1 shows the geographic and demographic characteristics of our sample in wave 1 and compares them to the 2020 ASEC data for respondents ages 60 to 68. The geographic distribution of our respondents surveys was similar to the 2020 ASEC sample as was the mean age, the shared married and the share divorced. Along several other dimensions, however, our respondents were more advantaged than the general population, with a higher share White, with a college or graduate degree, and a lower share Hispanic or uninsured. To the extent that changes in well-being are larger for the less advantaged, the estimates here will be a lower bound on the impact of the COVID-19 pandemic.

Mean self-rated health at baseline was nearly identical in our survey and the ASEC, although the distribution across ratings differed somewhat. We also compared the share with depressive symptoms (PHQ-2 $\geq$ 3) to those ages 60 to 68 in the 2018 MEPS. The share with depressive symptoms was quite similar– 7.2% in our survey versus 6.44% in the MEPS. The ladder was similar, if somewhat higher (better) in our sample (7.3) than in the U.S. population as a whole in 2018 (6.88) [26]. The proportion of individuals reporting pain yesterday in this sample (40%) was about 10 percentage points higher than the rate for this age range from a large, representative sample from the Gallup Organization using the same question [27].

Table 2 shows both the wave 1 and wave 2 mean for each of our six measures of well-being as well as the difference in these means and the p-value from a t-test of each difference. Between waves 1 and 2, the proportion of the sample with depressive symptoms increased from 7.24 to 8.65% or by 1.4 percentage points (95% CI, 0.97 to 1.86 percentage points) or about 20% relative to wave 1. Mean negative affect increased by about 0.23 (95% CI, 0.205 to 0.245) scale points off a base of about 1.1 scale points or 0.18 of the standard deviation of 1.25. Positive affect decreased by 0.10 (95% CI, -0.115 to -0.093) scale points off a base of 1.69 scale points or about 0.15 of the standard deviation of 0.66. The proportion reporting pain a lot of the day yesterday declined by 5 percentage points (95% CI, -5.8 to -4.3 percentage points) between waves 1 and 2, or nearly 13% off a base of 40% reporting a lot of pain in wave 1. The Cantril ladder decreased by 0.16 (95% CI, -0.183 to -0.137 scale points) off a base of 7.3 scale points or 0.09 of the standard deviation of 1.82 while self-rated health was unchanged.

In Fig 1, we show our estimates from Eq (1) of the wave 1 to wave 2 changes in well-being ($\alpha_1$) after controlling for fixed characteristics of respondents and stratified by sex, college degree, retirement status, race, marriage status and household income above versus below $50,000 at wave 1 (see S2 Table for estimates). Each horizontal bar represents an estimate from a different regression. Depressive symptoms were unchanged for men but increased for women by 2.4 percentage points (95% CI, 1.5 to 3.2 percentage points) or about 30% off a wave 1 rate of 8%. The increase in negative affect was larger for women (0.274 scale points; 95% CI 0.237 to 0.312) than men (0.148 scale points; 95% CI 0.106 to 0.191), for married (0.253 scale points; 95% CI 0.218 to 0.288) than unmarried respondents (0.175 scale points; 95% CI 0.127 to 0.223) and, somewhat unexpectedly, for those with household income at or above $50,000 at wave 1 (0.263 scale points; 95% CI 0.229 to 0.298) than those with income below that threshold (0.154 scale points; 95% CI 0.105 to 0.203). Changes in positive affect and pain did not differ by sex, education, retirement status, race, marriage or income status. As for the sample overall, self-rated health did not change for any of the sub-samples. The Cantril ladder declined (worsened) for all sub-groups, although the change was not statistically distinguishable from zero for the small sample of nonwhite respondents. The magnitude of the

**Table 1. Wave 1 Dynata sample characteristics[a] and comparison with the ASEC.**

| | Dynata, Wave 1 | 2020 CPS (ASEC) |
|---|---|---|
| | N = 16,644[b] | N = 16,382 |
| | Mean | Mean |
| Age | 64.3 | 63.8 |
| | (2.59) | (2.54) |
| New England | 5.6% | 4.9% |
| Mid-Atlantic | 15.3% | 12.8% |
| East North Central | 16.7% | 15.2% |
| West North Central | 6.8% | 6.9% |
| South Atlantic | 21.1% | 20.5% |
| East South Central | 4.4% | 6.3% |
| West South Central | 7.9% | 11.4% |
| Mountain | 8.0% | 7.2% |
| Pacific | 12.9% | 14.8% |
| White | 91.1% | 80.9% |
| Non-white | 8.9% | 19.1% |
| Hispanic | 3.1% | 10.4% |
| Female | 61.1% | 52.3% |
| Married | 64.6% | 65.0% |
| Divorced | 16.5% | 16.1% |
| HS or below | 15.9% | 39.4% |
| some college | 20.1% | 16.1% |
| college (AA + BA) | 43.2% | 31.3% |
| graduate deg | 20.8% | 13.2% |
| working | 34.1% | 45.0% |
| income less than $50,000 | 34.9% | 33.0% |
| income of $50,000 or more | 65.1% | 67.0% |
| Uninsured | 2.4% | 5.0% |
| Self-rated health | 2.57 | 2.58 |
| | (0.92) | (1.09) |
| Depressive symptoms[c] | 7.2% | 6.44% |
| Cantril ladder | 7.30 | |
| | (1.82) | |
| Negative Affect | 1.09 | |
| | (1.25) | |
| Positive Affect | 1.69 | |
| | (0.66) | |
| Pain Yesterday | 40.0% | |

Notes

[a] Wave 1 data are from an internet survey of a panel of respondents from Dynata and were collected between November 2019 and February 2020.

[b] Outcome measures from the Dynata survey are missing for a few respondents and vary from 16,640 respondents for positive affect to 16,644 for pain.

[c] Data on depressive symptoms are from the 2018 Medical Expenditure Panel Survey (MEPS) and are based on 2,901 respondents ages 60 to 68.

**Table 2. Measures of well-being across survey waves 1 and 2[a].**

| Variable | Number of Respondents | Wave 1 Mean (std dev) | Wave 2 Mean (std dev) | Difference across Waves | p-value[c] |
|---|---|---|---|---|---|
| Depressive symptoms | 16,641 | 7.24% | 8.65% | 1.42p.p.[b] | <0.001 |
| Negative affect index | 16,639 | 1.09 | 1.31 | 0.225 | <0.001 |
|  |  | (1.25) | (1.32) |  |  |
| Positive affect index | 16,639 | 1.69 | 1.58 | -0.104 | <0.001 |
|  |  | (0.66) | (0.74) |  |  |
| Pain | 16,642 | 40.0% | 34.9% | -5.1 p.p. | <0.001 |
| Cantril ladder | 16,633 | 7.30 | 7.14 | -0.160 | <0.001 |
|  |  | (1.82) | (1.83) |  |  |
| Self-rated Health | 16,644 | 2.57 | 2.58 | 0.004 | 0.379 |
|  |  | (0.92) | (0.91) |  |  |

Notes

[a] Data are from two waves of an internet survey of a panel of respondents from Dynata. Wave 1 data were collected between November 2019 and February 2020. Wave 2 data were collected between April and May 2020.

[b] p.p. denotes percentage points.

[c] This is the p-value from a paired test of the difference in means across waves.

change in the Cantril ladder varied little across demographic groups with the exception of marriage status and income: the ladder declined by 0.209 (95% CI: -0.247 to -0.170) scale points or about about 0.12 of the standard deviation of 1.6 for married respondents but only 0.072 (95% CI: -0.132 to -0.012) or 0.04 of the standard deviation of 1.7 for unmarried respondents. While we found no change in the Cantril ladder for those with income below $50,000 at wave 1 (-0.028 scale points; 95% CI -0.092 to 0.035), the ladder declined by 0.231 scale points for those with income above $50,000 at wave 1 (95% CI: -0.268 to -0.194) or 0.15 of the standard deviation of 1.55 for the higher income group.

Fig 2 displays estimates of how living in counties above relative to below the 90th percentile of COVID-19 death rates, rating yourself above relative to below the median likelihood of (1) contracting COVID-19 and, separately (2) dying from COVID-19, whether COVID-19 kept you from exercising or resulted in loss of job/income and whether the respondent was female, had no college education, was married (versus unmarried), nonwhite or had income below (relative to at or above) $50,000, were jointly related to the across-wave z-score of changes in outcomes (see S3 Table for all model estimates). All points in a given panel represent estimates from a single regression (vectors $\Theta$ and $\beta$ in Eq 2), reflecting partial correlations controlling for the other variables in the model.

Our results show that the changes in well-being documented above were strongly related to local COVID-19 death rates, to the pandemic's effects on respondends routines and livelihoods, and to respondends' subjective perceptions of potential COVID-19 risks to their own health. In addition to sex (female), the increase in the rate of depressive symptoms was larger for individuals who were above the median in rating their chance of contracting COVID-19 (0.033 of a standard deviation (s.d.); 95% CI, 0.002 to 0.064), above median in rating their chance of dying from the disease conditional on getting it (0.057 of a s.d.; 95% CI, 0.024 to 0.089) and who reported being kept from exercising by COVID-19 (0.097 s.d.; 95% CI, 0.062 to 0.131).

Negative affect increased more for respondents in counties with COVID-19 death rates above versus below the 90th percentile (0.097 s.d.; 95% CI 0.042 to 0.152) as well as among individuals above the median in rating their chance of getting COVID-19 (0.132 s.d.; 95% CI 0.096

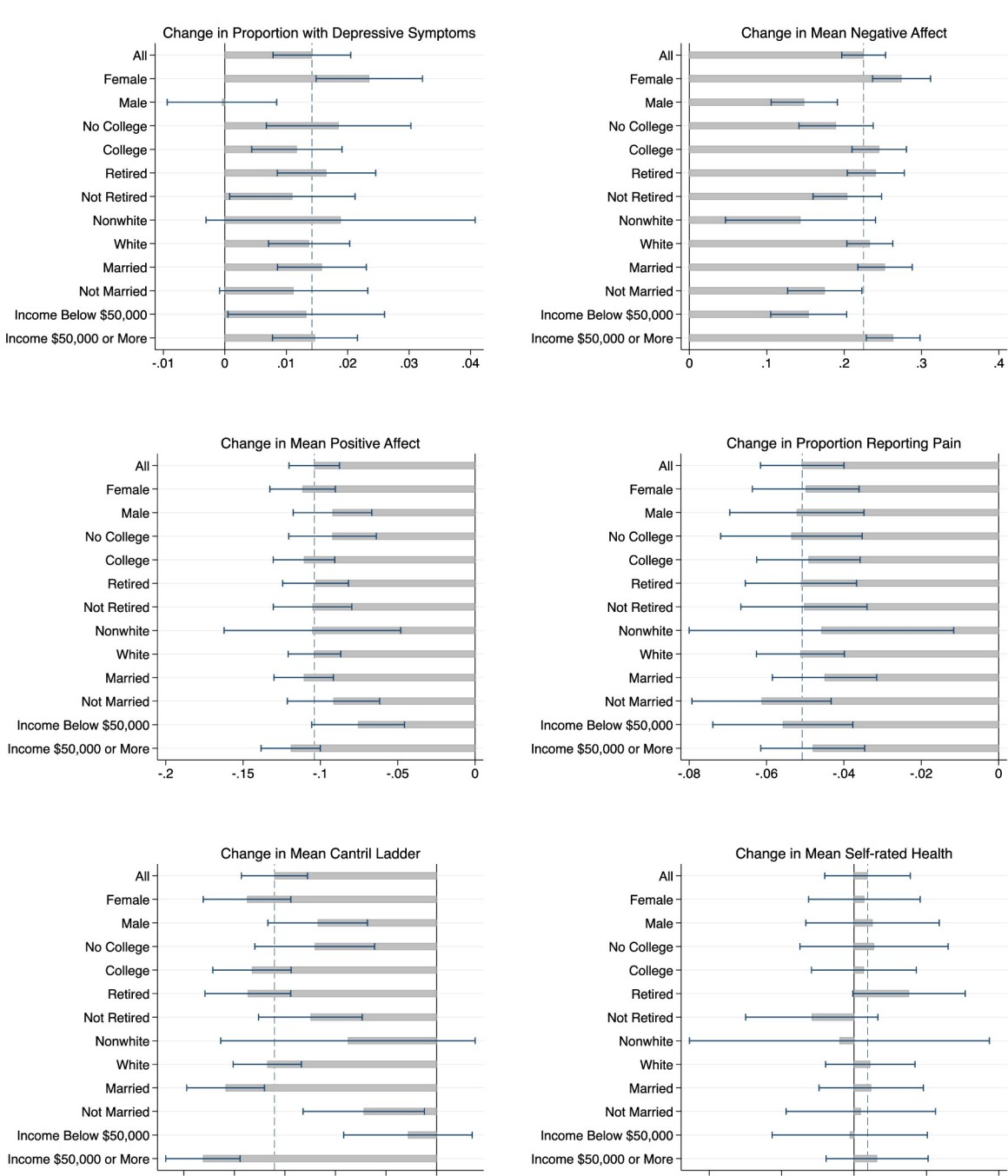

**Fig 1. Changes in well-being by respondent characteristics.** Notes: Each bar represents the wave 1 to wave 2 change, based on Eq (1), in the outcome for the specified sample–all respondents, females only, males only, those without a college degree, those with a college degree or higher, those who were retired at wave 1, those who were not retired at wave 1, those who are white, those who are non-white, those with household income below $50,000, those with household income at or above $50,000. The line at 0 denotes no change; the dashed blue line denotes the change for the overall sample.

to 0.133) and dying from the disease conditional on getting it (0.099 s.d.; 95% CI 0.065 to 0.133) and who reported being kept from exercising by COVID-19 (0.079 scale points; 95% CI 0.043 to 0.114). Married respondents and those who were retired at wave 1 also experienced larger increases in positive affect while those with income below $50,000 had small increases.

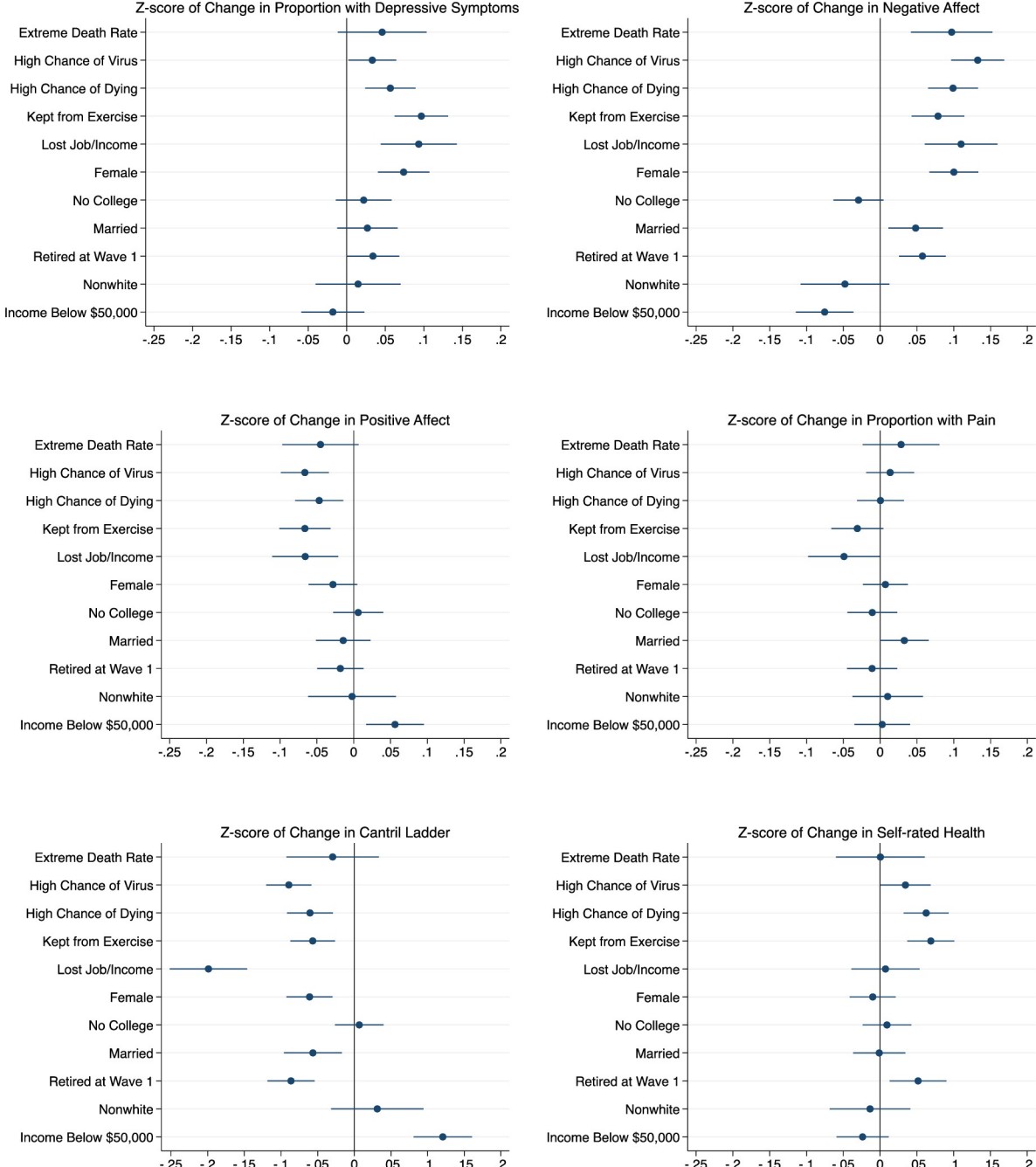

**Fig 2. Impact of demographics and COVID-19 factors on changes in well-being.** Notes: Estimates are from the z-score of first-difference regression models specified in Eq (2)."Extreme Death Rate" means the respondent lived in a county that was in the 90[th] percentile of COVID-19 death rates the day before the respondent answered the wave 2 survey. "High Chance of Virus" means the respondent rated their odds of contracting the virus above the median respondent. "High Chance of Dying" means that once infected, the respondent rated their odds of dying from the virus above the median respondent.

Positive affect decreased more for respondents who were above the median in rating their chance of getting COVID-19 (-0.066 s.d.; 95% CI -0.099 to -0.034) and dying from the disease conditional on getting it (-0.047 scale points; 95% CI -0.080 to -0.014) and who reported being kept from exercising by COVID-19 (-0.066 scale points; 95% CI -0.080 to -0.014). The

decrease was smaller for those with income below $50,000 in wave 1 (0.056 s.d.; 95% CI 0.017 to 0.095).

The decline in pain was generally unrelated to the demographic and pandemic related factors measured here. One exception was that the decline in pain was larger by -0.049 s.d. (95% CI, -0.098 to -0.000 percentage points) for those who reported losing their job or a significant amount of income due to the pandemic.

Although we found no change overall, self-rated health worsened (increased) by 0.034 s.d. (95% CI, 0.000 to 0.069) for those respondents above the median in rating their chance of getting COVID-19, by 0.063 s.d. (95% CI 0.032 to 0.093) for those above the median in rating their chance of dying from the virus if infected and by 0.069 s.d. (95% CI 0.037–0.101) for those kept from exercise due to COVID-19. These same factors were also associated with a worsening (decline) in the Cantril ladder. In addition, we see that losing a job or income due to COVID-19 has a large negative effect on the Cantril ladder, -0.199 (95% CI -0.252 to -0.146). Changes in the Cantril ladder were also larger for those who were married -0.056 s.d. (95% CI -0.096 to -0.017), and retired at wave 1, -.086 s.d. (95% CI -0.118 to -0.054). The decrease in the Cantrill ladder was smaller for those with income below $50,000 in wave 1, 0.121 s.d. (95% CI 0.081 to 0.161).

## Discussion

Our results suggest a complex pattern of short-run changes in the well-being of seniors just before and after COVID-19 was recognized as a public health crisis in the United States. Depressive symptoms, negative affect and positive affect worsened across the two survey waves. The Cantril ladder changed modestly while self-reported health was unchanged and pain actually improved.

The increase in rates of depressive symptoms and negative affect and decrease in positive affect were expected and tended to vary in expected ways with COVID-related factors such as loss of a job/income due to the pandemic, fears of getting sick or dying from COVID-19 and limits on the ability to exercise as a result of the virus, perhaps due to closed gyms or fears of going outside.

Consistent with a hypothesized worsening of evaluative well-being, the Cantril ladder declined. With the exception of a few sub-groups such as married respondents and those retired at wave 1, the decline was small. The Cantril ladder also varied with COVID-related factors, decreasing more for those who lost their job or income due to the pandemic or thought they had a high chance of dying from the disease. Self-rated health was unchanged across waves. One possibility for these limited changes is that these outcomes capture more stable, long-run measures of well-being whereas affect and depressive symptoms, which refer to yesterday and the past 7-days respectively, assess feelings over the short-run and may be more likely to capture recent changes due to the panademic. In addition, respondents might think in terms of relative well-being when answering evaluative measures but not affect measures. The worsening of negative affect, positive affect and the Cantril ladder was smaller for those with income below $50,000 at wave 1. The reasons for this are unclear and are the subject of ongoing work by this team.

The sizeable decline in pain, which also captures experience yesterday, is more puzzling. While the rate of pain declined more for those who reported losing their job or significant income due to COVID-19, it did not vary by our survey measures of COVID-19 related beliefs, changes in exercise, or living in high COVID-19 mortality areas. That the decline was larger for those with reduced economic activity suggests it may be related to reduced everyday bodily wear and tear, although our current data are not well suited to assessing this issue.

## Limitations

This study has several limitations. First, the survey was conducted on-line, which may bias the results towards respondents with access to the internet and facility using internet-enabled devices. Second, survey attrition was more common among the less advantaged, which could lead us to understate changes in well-being to the extent they were larger among those already facing hardship. Third, wave 1 data were conducted over a 4 month period and may capture temporal changes in well-being, including changes related to the spread of COVID-19 globally. Fourth, our data focused on younger seniors, all of whom lived independently, and could not speak to the experience of older seniors or those living in institutional settings, where loneliness and isolation due to COVID-19 protocols may have had different impacts. Finally, since the pandemic affected everyone in the United States in some way, we do not have a control group that captures how well-being would have changed absent COVID-19. That said several factors suggest that absent the pandemic well-being would have improved. In particular, in the 2018 MEPS we find declining rates of depressive symptoms from January to May (see S5 Table). Likewise, the age-profile of negative and positive affect [28] as well as the Cantril ladder [29], which improves with age after about age 50, and seasonal changes in depressive symptoms, which some work finds ameliorates in the spring [30], both suggest that absent COVID-19 well-being would have improved across survey waves (wave 1 was conducted in November-February and wave 2 from April-May). Thus, our estimates will, if anything, understate any negative impact of COVID-19 on depressive symptoms and affect.

## Conclusions

Consistent with prior evidence, we find that depressive symptoms and both negative and positive affect worsened during the pandemic. However, our work also suggests that the assumption of exclusively negative changes in well-being may be unfounded. At least for the young seniors in our data, COVID-19 did not markedly change self-rated health or evaluative well-being, and pain improved. This does not diminish the importance of the increases in depressive symptoms and negative affect or the decrease in positive affect but rather suggests that a more holistic view of well-being may be warranted. Moreover, as the COVID-19 pandemic continues to disrupt life in the United States, evaluative well-being and self-rated health may change in ways similar to depressive symptoms and affect. Likewise, the reduction in pain, if attributable to reduced activity, could reverse in the long-run as sedentary lifestyles can increase pain.

## Supporting information

**S1 File. Wave 1 survey.**
(DOCX)

**S2 File. Wave 2 survey 2.**
(DOCX)

**S3 File. Analytic dataset.**
(DTA)

**S4 File. Other supporting datasets.**
(ZIP)

**S5 File. Replication code.**
(ZIP)

**S6 File. Information sheet.**
(DOCX)

**S1 Table. Predictors of only wave 1 participation.**
(PDF)

**S2 Table. Changes in well-being across waves by demographic characteristics.**
(PDF)

**S3 Table. Predictors of the Z-score of changes in well-being across waves.**
(PDF)

**S4 Table. Predictors of changes in well-being across waves.**
(PDF)

**S5 Table. Depressive symptoms in Jan/Feb vs. Apr/May in the 2018 medical expenditure panel survey.**
(PDF)

## Acknowledgments

Andrew Yu provided excellent research assistance.

## Author Contributions

**Conceptualization:** Silvia Barcellos, Mireille Jacobson.

**Formal analysis:** Mireille Jacobson.

**Funding acquisition:** Silvia Barcellos, Mireille Jacobson.

**Project administration:** Silvia Barcellos.

**Visualization:** Arthur A. Stone.

**Writing – original draft:** Mireille Jacobson.

**Writing – review & editing:** Silvia Barcellos, Arthur A. Stone.

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
