## [Decision Letter · Decision Letter 0]

12 Mar 2021

PONE-D-20-40269

Varied and Unexpected Changes in the Well-being of Seniors in the United States amid the COVID-19 Pandemic

PLOS ONE

Dear Dr. Mireille Jacobson,

Thank you for submitting your manuscript to PLOS ONE. After careful consideration, we feel that it has merit but does not fully meet PLOS ONE’s publication criteria as it currently stands. Therefore, we invite you to submit a revised version of the manuscript that addresses the points raised during the review process.

We look forward to receiving your revised manuscript.

Kind regards,

Yuka Kotozaki

Academic Editor

PLOS ONE

Journal Requirements:

2.Please provide additional details regarding participant consent. In the ethics statement in the Methods and online submission information, please ensure that you have specified (1) whether consent was informed and (2) what type you obtained (for instance, written or verbal, and if verbal, how it was documented and witnessed). If your study included minors, state whether you obtained consent from parents or guardians. If the need for consent was waived by the ethics committee, please include this information.

3. In the Methods section and the online submission, please further clarification whether the original study, conducted between November 2019 and February 2020 received ethical approval and please provide the ethical approval number.

4. Please include additional information regarding the survey or questionnaire used in the study and ensure that you have provided sufficient details that others could replicate the analyses. For instance, if you developed a questionnaire as part of this study and it is not under a copyright more restrictive than CC-BY, please include a copy, in both the original language and English, as Supporting Information.

5.In your Data Availability statement, you have not specified where the minimal data set underlying the results described in your manuscript can be found. PLOS defines a study's minimal data set as the underlying data used to reach the conclusions drawn in the manuscript and any additional data required to replicate the reported study findings in their entirety. All PLOS journals require that the minimal data set be made fully available. For more information about our data policy, please see http://journals.plos.org/plosone/s/data-availability.

6.We note that the grant information you provided in the ‘Funding Information’ and ‘Financial Disclosure’ sections do not match.

7. Please include your tables as part of your main manuscript and remove the individual files. Please note that supplementary tables (should remain/ be uploaded) as separate "supporting information" files

Reviewers' comments:

Reviewer's Responses to Questions

**Comments to the Author**

1. Is the manuscript technically sound, and do the data support the conclusions?

Reviewer #1: Yes

2. Has the statistical analysis been performed appropriately and rigorously? 

Reviewer #1: Yes

3. Have the authors made all data underlying the findings in their manuscript fully available?

Reviewer #1: Yes

4. Is the manuscript presented in an intelligible fashion and written in standard English?

Reviewer #1: Yes

5. Review Comments to the Author

Reviewer #1: PLOS ONE

Varied and Unexpected Changes in the Well-Being of Seniors in the United States

amid the COIVD-19 Pandemic

The authors examine and present changes in several measures of well-being collected in 2-waves of a national, online panel survey of seniors aged 60-68, before and during the COVID-19 pandemic. In the Introduction, the authors note that prior studies of pandemic-induced changes in well-being of older adults have relied on cross-sectional data or follow respondents post-pandemic. Thus, the significance of the study derives from the panel study design, allowing assessments of well-being before versus after the start of the pandemic. Most comments below request more information/details about the study.

1. Perhaps the most important comment is that the Introduction section lacks information about the context of the pandemic in the U.S. in 2020. Contextualizing the pandemic is important because context is the basis for (or the reasons) why older adults (and other age groups) might be worse off after the pandemic, compared to before the pandemic. Salient elements of context include, but are not limited to, societal changes as the shutdown of the U.S. economy, stay-at-home orders, working from home (when possible), wearing masks, social distancing, closing schools, reduced access to grandchildren and aging parents in nursing homes, and so on. All of these, taken together, are compelling contextual reasons why people’s well-being might decline during the pandemic. Also, five or ten years from now, readers’ memories of the pandemic’s societal impacts may have faded; adding information about context will be a useful reminder to future readers.

Also, related to the above comment, while the panel study design is great, there is no control group: there is no geographic area of the U.S. that is not exposed to the coronavirus. Therefore, it is impossible to estimate how much well-being might have changed in the absence of the pandemic. Does the literature indicate that the secular trend for this age group is one of increasing or decreasing well-being over time, or stable well-being over time?

2. In the Introduction section on page 3, there is a sentence noting that past studies have found that seniors have fared better than other age groups during the pandemic. Why is this so?

3. In the Introduction, the authors hypothesize that changes in well-being may vary by baseline characteristics, such as gender and education. What is the basis (or reasons) for the hypothesized variation?

4. The following comments are for the Methods section:

The baseline survey runs from November 2019 to February 2020. The second wave survey began in April 2020. When did the second wave survey end?

Page 4 indicates respondents were matched to the COVID-19 death count in their county of residence at wave 2. However, in the Results section, findings are presented for COVID-19 rates rather than counts.

In the measures section, please clarify whether the pain measure is for physical pain. On the Web, the documentation for the Cantril ladder indicates the top of the ladder is defined as the best possible life, and the bottom of the ladder is defined as the worst possible life. Consider noting those anchors in the text.

In the regression models with the COVID-19 variables, did those regression models include fixed effects, like in the first set of models?

5. The following comments are for the Results section:

On page 7, the authors describe the change in negative affect off the base. Please add the base value for positive affect and for the Cantril ladder. About 40% of respondents reported pain in wave 1, which sounds high – perhaps there was regression toward the mean?

In Table 1, why are the sample sizes smaller for depressive symptoms and pain?

In Table 1, please consider adding a column on the far right reporting the differences in the means, and whether the differences are significant. Is the statistical test for differences in wave 1 versus wave 2 a bivariate test (unadjusted)? If t-tests are conducted, were paired t-tests conducted?

Table 1 just contains descriptive statistics for the dependent variables. Please add a table indicating descriptive statistics for sociodemographic characteristics in wave 1.

In the footnotes to Figure 1 and Figure 2, please add a sentence describing the regression models that produced the estimates in each Figure.

There are increasing reports in the literature that the pandemic is having more severe impacts on vulnerable groups. In Figure 1, does the data set allow comparison of low-come versus other incomes? Comparisons of other racial groups within the broad “nonwhite” category?

6. In the Discussion section, the authors note that self-rated health did not change between wave 1 and wave 2. Does this finding suggest that general measures of health status are less susceptible to change than the more specific measures of emotional well-being?

6. PLOS authors have the option to publish the peer review history of their article (what does this mean?). If published, this will include your full peer review and any attached files.

Reviewer #1: **Yes: **David Grembowski

---

## [Author Response · Author response to Decision Letter 0]

22 Apr 2021

Please see the response document included.

---

## [Decision Letter · Decision Letter 1]

9 May 2021

PONE-D-20-40269R1

Varied and Unexpected Changes in the Well-being of Seniors in the United States amid the COVID-19 Pandemic

PLOS ONE

Dear Dr. Mireille Jacobson,

Thank you for submitting your manuscript to PLOS ONE. After careful consideration, we feel that it has merit but does not fully meet PLOS ONE’s publication criteria as it currently stands. Therefore, we invite you to submit a revised version of the manuscript that addresses the points raised during the review process.

We look forward to receiving your revised manuscript.

Kind regards,

Yuka Kotozaki

Academic Editor

PLOS ONE

Journal Requirements:

Reviewers' comments:

Reviewer's Responses to Questions

**Comments to the Author**

1. If the authors have adequately addressed your comments raised in a previous round of review and you feel that this manuscript is now acceptable for publication, you may indicate that here to bypass the “Comments to the Author” section, enter your conflict of interest statement in the “Confidential to Editor” section, and submit your "Accept" recommendation.

Reviewer #1: (No Response)

2. Is the manuscript technically sound, and do the data support the conclusions?

Reviewer #1: Yes

3. Has the statistical analysis been performed appropriately and rigorously? 

Reviewer #1: Yes

4. Have the authors made all data underlying the findings in their manuscript fully available?

Reviewer #1: Yes

5. Is the manuscript presented in an intelligible fashion and written in standard English?

Reviewer #1: Yes

6. Review Comments to the Author

Reviewer #1: PLOS ONE

Varied and Unexpected Changes in the Well-Being of Seniors in the United States

amid the COIVD-19 Pandemic

Revision

The authors examine and present changes in several measures of well-being collected in 2-waves of a national, online panel survey of seniors aged 60-68, before and during the COVID-19 pandemic. The revised manuscript is responsive to comments raised in the prior original manuscript. Some minor comments are presented below for consideration.

1. The Introduction section is very engaging and should be well-received by readers. A minor point is that, in the last sentence of the Introduction, the authors note that COVID-19 may have differential effects on the study’s dependent variables, which are likely mediated by education, gender, race and other pre-determined characteristics. This sentence may trigger impressions, in the minds of readers, that the authors will be doing a mediation analysis, which is not the case. Some options might be to change this sentence, or perhaps retain the sentence, as is, but touch on this “mediation theme” in the Discussion section.

2. In the Measures section, for clarity, please note the range of the negative affect scale and the range of the positive affect scale.

3. For clarity, in the Statistical Analysis section, please add a sentence defining the terms in Equation 1.

4. The Results section begins with comparisons of the study’s survey characteristics with national representative data sets. These comparisons are great. However, the Methods section does not indicate that such comparisons will be performed.

5. In Table 1, the authors and editor might consider simply noting the sample sizes a single time at the top of the columns.

But please note the 2901 MEPS sample size for depression in the (b) footnote.

6. Could the decline in pain be regression to the mean?

7. Some of the dependent variables unexpectedly declined more for people with incomes above $50,000. Could this be because the pandemic led to a shutdown in the economy and increased unemployment and/or reduced hours worked per week, and people in the above $50,000 category had “more to lose” than people who had low incomes pre-pandemic – resulting in greater declines in the Cantril ladder well-being scores for the higher income group (?).

7. PLOS authors have the option to publish the peer review history of their article (what does this mean?). If published, this will include your full peer review and any attached files.

Reviewer #1: **Yes: **David Grembowski

---

## [Author Response · Author response to Decision Letter 1]

10 May 2021

Please see included response document.

---

## [Decision Letter · Decision Letter 2]

26 May 2021

Varied and Unexpected Changes in the Well-being of Seniors in the United States amid the COVID-19 Pandemic

PONE-D-20-40269R2

Dear Dr. Mireille Jacobson,

We’re pleased to inform you that your manuscript has been judged scientifically suitable for publication and will be formally accepted for publication once it meets all outstanding technical requirements.

Kind regards,

Yuka Kotozaki

Academic Editor

PLOS ONE

Additional Editor Comments (optional):

Reviewers' comments:

Reviewer's Responses to Questions

**Comments to the Author**

1. If the authors have adequately addressed your comments raised in a previous round of review and you feel that this manuscript is now acceptable for publication, you may indicate that here to bypass the “Comments to the Author” section, enter your conflict of interest statement in the “Confidential to Editor” section, and submit your "Accept" recommendation.

Reviewer #1: All comments have been addressed

2. Is the manuscript technically sound, and do the data support the conclusions?

Reviewer #1: Yes

3. Has the statistical analysis been performed appropriately and rigorously? 

Reviewer #1: Yes

4. Have the authors made all data underlying the findings in their manuscript fully available?

Reviewer #1: Yes

5. Is the manuscript presented in an intelligible fashion and written in standard English?

Reviewer #1: Yes

6. Review Comments to the Author

Reviewer #1: (No Response)

7. PLOS authors have the option to publish the peer review history of their article (what does this mean?). If published, this will include your full peer review and any attached files.

Reviewer #1: **Yes: **David Grembowski

---

## [Editor Report · Acceptance letter]

10 Jun 2021

PONE-D-20-40269R2 

Varied and Unexpected Changes in the Well-being of Seniors in the United States amid the COVID-19 Pandemic 

Dear Dr. Jacobson:

I'm pleased to inform you that your manuscript has been deemed suitable for publication in PLOS ONE. Congratulations! Your manuscript is now with our production department. 

Kind regards, 

on behalf of

Dr. Yuka Kotozaki 

Academic Editor

PLOS ONE